# Differentiating High-Grade Gliomas from Brain Metastases at Magnetic Resonance: The Role of Texture Analysis of the Peritumoral Zone

**DOI:** 10.3390/brainsci10090638

**Published:** 2020-09-16

**Authors:** Csaba Csutak, Paul-Andrei Ștefan, Lavinia Manuela Lenghel, Cezar Octavian Moroșanu, Roxana-Adelina Lupean, Larisa Șimonca, Carmen Mihaela Mihu, Andrei Lebovici

**Affiliations:** 1Radiology and Imaging Department, County Emergency Hospital, Cluj-Napoca, Clinicilor Street, Number 5, Cluj-Napoca, 400006 Cluj, Romania; csutakcsaba@yahoo.com (C.C.); pop.lavinia@umfcluj.ro (L.M.L.); carmenmihu@umfcluj.ro (C.M.M.); andrei1079@yahoo.com (A.L.); 2Radiology, Surgical Specialties Department, “Iuliu Haţieganu” University of Medicine and Pharmacy, Clinicilor Street, number 3–5, Cluj-Napoca, 400006 Cluj, Romania; 3Anatomy and Embryology, Morphological Sciences Department, “Iuliu Haţieganu” University of Medicine and Pharmacy, Victor Babeș Street, number 8, Cluj-Napoca, 400012 Cluj, Romania; 4Department of Neurosurgery, North Bristol Trust, Southmead Hospital, Southmead Road, Westbury on Trym, Bristol BS2 8BJ, UK; cezar.morosanu@nbt.nhs.uk; 5Histology, Morphological Sciences Department, “Iuliu Hațieganu” University of Medicine and Pharmacy, Louis Pasteur Street, number 4, Cluj-Napoca, 400349 Cluj, Romania; roxanalupean92@gmail.com; 6Department of Paediatric Surgery, Bristol Royal Hospital for Children, Upper Maudlin Street, Bristol BS2 8BJ, UK; larisa.simonca@UHBristol.nhs.uk

**Keywords:** glioblastoma, computer-aided diagnosis, magnetic resonance imaging, texture analysis

## Abstract

High-grade gliomas (HGGs) and solitary brain metastases (BMs) have similar imaging appearances, which often leads to misclassification. In HGGs, the surrounding tissues show malignant invasion, while BMs tend to displace the adjacent area. The surrounding edema produced by the two cannot be differentiated by conventional magnetic resonance (MRI) examinations. Forty-two patients with pathology-proven brain tumors who underwent conventional pretreatment MRIs were retrospectively included (HGGs, *n* = 16; BMs, *n* = 26). Texture analysis of the peritumoral zone was performed on the T2-weighted sequence using dedicated software. The most discriminative texture features were selected using the Fisher and the probability of classification error and average correlation coefficients. The ability of texture parameters to distinguish between HGGs and BMs was evaluated through univariate, receiver operating, and multivariate analyses. The first percentile and wavelet energy texture parameters were independent predictors of HGGs (75–87.5% sensitivity, 53.85–88.46% specificity). The prediction model consisting of all parameters that showed statistically significant results at the univariate analysis was able to identify HGGs with 100% sensitivity and 66.7% specificity. Texture analysis can provide a quantitative description of the peritumoral zone encountered in solitary brain tumors, that can provide adequate differentiation between HGGs and BMs.

## 1. Introduction

High-grade gliomas (HGGs) are the most common malignant brain tumors in adults, while brain metastases (BMs) affect 20–30% of patients with cancer [1]. The clinical and imaging differentiation between the two entities is crucial, as they have very different management strategies [2].

Tumor angiogenesis with disruption of the blood–brain barrier (BBB) is responsible for the vasogenic edema encountered in both benign and malignant brain lesions. Two types of vasogenic peritumoral edema have been defined. The first type is due to parenchymal compression which can lead to secondary ischemia and can be found near low-grade and nonglial tumors [3]. Because there is no histological evidence of tumoral cellularity outside the margins of metastatic lesions, this edema is considered “pure vasogenic” [1]. The second type is encountered in high-grade glial tumors and is caused by additional derangements of the BBB by malignant cell infiltration [3]. Thus, it is possible that an accurate imaging assessment of the malignant cell infiltration within the peritumoral edema can provide a noninvasive diagnostic alternative in distinguishing HGGs from brain metastases. Conventional magnetic resonance imaging (MRI) examination often shows common features of solitary metastatic brain tumors and high-grade gliomas [4,5], as they both express surrounding edema, necrotic center, and irregular enhancing margins [6]. These similarities lead to imaging misclassification in more than 40% of cases [7]. Because BMs tend to displace rather than invade surrounding tissues, the peritumoral edema is considered pure vasogenic [8,9]. On the other hand, HGGs and especially glioblastomas usually invade the surrounding structures, tumor cells infiltrating the white matter [10]. An important key point in discriminating HGGs from BMs may lie in detecting the differences between the two types of edemas.

Although conventional MRI brain examinations are widely available, the standard sequences lack pathology-specific biomarkers and the signal does not possess biological specificity. Thus, a nonspecific increase in the blood–brain barrier permeability is reflected by contrast enhancement, and the tissue water content influences T2-weighted signal abnormalities. Due to these factors, conventional MRI is unable to accurately characterize the surrounding edema of solitary brain tumors [11]. More advanced MRI techniques such as spectroscopy, perfusion, diffusion-weighted, and diffusion tensor imaging have also been used to quantify the peritumoral zone, but the results are often contradictory [10,12,13].

Most certainly, standard MRI sequences carry additional diagnostic information, but this information is difficult to be assessed during the routine evaluation of medical images. The histopathological particularities of the two types of edema may influence the pixel intensity and spatial distribution within MRI images, but these changes are too subtle to be macroscopically quantified.

Computer-aided diagnostic methods involving medical images have advanced in recent years from a small research topic to a successful implementation in some areas of clinical practice [14]. Many of these augmented diagnostic methods rely on the evaluation of textures and promise major upgrades in the way physicians interpret examinations [15]. Textures represent the intrinsic and intuitive properties of surfaces such as roughness, granulation, and regularity [16]. In recent years, texture analysis (TA) has emerged as a noninvasive method for quantifying the information composed especially in magnetic resonance images [17]. TA represents a noninvasive radiomics method to assess macroscopic tissue heterogeneity which is indirectly linked to microscopic tissue heterogeneity indiscernible to human visual perception [18]. Through specific parameters, TA aims to offer a quantitative assessment of image contents by analyzing the distribution patterns and intensity of the pixels [19]. So far, most TA studies involving gliomas have focused on solid tumor components [20,21], while the surrounding peritumoral aspects remained relatively unexplored.

Our objective was to analyze the peritumoral zone of solitary intracerebral brain lesions with texture analysis to determine noninvasive differentiation criteria for HGGs and BMs. We aimed to assess the diagnostic power of texture parameters by evaluating their individual and combined capability of discriminating between the two entities.

## 2. Materials and Methods

### 2.1. Patients

This Health Insurance Portability and Accountability Act-compliant, single-institution, retrospective pilot study was approved by the institutional review board (ethics committee of the “Iuliu Hațieganu” University of Medicine and Pharmacy Cluj-Napoca; registration number, 50; date, 11 March 2019), and a waiver consent was obtained owing to its retrospective nature. The database was obtained from our radiology information system (RIS) and consisted of all the reports of MRI brain examinations performed between May 2016 and March 2020. The original search yielded 1765 reports. Each report was analyzed, and those reports that did not refer to a supratentorial intraparenchymal brain mass were excluded (*n* = 942), as well as those that refer to the recurrence of a malignant lesion (*n* = 196). The medical records of the remaining 627 patients were retrieved from the archive of our healthcare unit to be investigated for disease-related data. Further exclusion criteria were: the benign nature of a lesion (*n* = 263), the absence of a final histopathological result (*n* = 84), and nonsolitary lesions (*n* = 177). The remaining 103 studies were reviewed by one radiologist, which excluded all examinations with artifacts affecting the quality of the T1- and T2-weighted images (T2WI, *n* = 19), tumors below 25 mm (*n* = 61), and lesions with a maximum diameter of the peritumoral edema below 15 mm as visible on fluid-attenuated inversion recovery (FLAIR) sequence (*n* = 23; Figure 1).

Of the 1765 patients who referred to our department during the study period, 42 were included in the study after applying the inclusion and exclusion criteria (29 men, 17 women; mean age = 62.6 ± 12.3 years; age range = 32–81 years). Subjects were further divided according to the final pathological diagnosis of their lesions in high-grade gliomas (HGGs, *n* = 16) and brain metastases (BMs, *n* = 26). Gliomas were classified according to the 2016 World Health Organization (WHO) classification of tumors of the central nervous system. Out of 42 patients, WHO grade IV glioblastoma was diagnosed in 11 patients (six males (m)/5 females (f)) and WHO grade III glioma in five patients (three anaplastic astrocytomas (2m/1f) and two anaplastic oligoastrocytomas (2m)). The brain metastases group included: 13 subjects with pulmonary adenocarcinoma (13m/4f), four with melanoma (2m/2f), three patients with colorectal carcinoma (1m/2f), two with pulmonary squamous cell carcinoma (2m), two with breast cancer (2f), one with a neuroendocrine tumor of the mesentery, (1f) and one with clear cell renal carcinoma (1m).

### 2.2. MRI Protocol

All scans were performed on the same unit (1.5-T, SIGNA™ Explorer, General Electric, GE Healthcare, Fairfield, Waukesha, WI, USA). Conventional sequences, including T1-weighted images, T2-weighted images, fluid-attenuated inversion recovery images, diffusion-weighted images, and contrast-enhanced T1-weighted images were obtained. Although the protocol varied because the examinations were retrieved from a range of approximately 4 years, each protocol included an axial T2-weighted Periodically Rotated Overlapping Parallel Lines with Enhanced Reconstruction (PROPELLER) sequence (repetition time, 7300 ms; echo time, 107 ms; slice thickness, 4 mm; field of view, 240 × 240 mm^2^; matrix, 384 × 384), and a contrast-enhanced T1-weighted fast spin echo (FSE) sequence (repetition time, 9 ms; echo time, 2.38 ms; slice thickness, 2.2 mm; field of view, 180 × 240 mm^2^; matrix, 256 × 256), which were the only ones used for the texture computation.

### 2.3. Texture Analysis Protocol

The traditional approach of radiomics consists of four steps: image segmentation using regions of interest, feature extraction, feature selection, and prediction. On a dedicated workstation (General Electric, Advantage workstation, 4.7 edition), all examinations were reviewed by two radiologists, blinded to the final diagnosis. The two examiners develop a common opinion about the slice considered to be the most representative for the peritumoral region from the postcontrast T1-weighted (T1W) sequence. The selected slice, along with the corresponding image on the T2-weighted (T2W) sequence, were retrieved in Digital Imaging and Communications in Medicine (DICOM) format and further imported in a texture analysis software, MaZda version 5 (Institute of Electronics, Technical University of Lodz, Łódź, Poland) [22]. The area outside the solid part of the tumor was defined as the peritumoral region. On the postcontrast T1W reference images, a region of interest (ROI) was placed as closely as possible to the tumoral margin. A semiautomatic level set technique was used for the definition and positioning of each ROI. One researcher placed a seed near the tumoral boundaries, and the software automatically delineated the peritumoral zone based on gradient and geometric coordinates. Since this technique did not involve manual delimitation of the structure of interest contours, the assessment of inter- or intraobserver reproducibility was not performed in the current study. Once defined, the ROI was transferred to the corresponding T2W slice, on which texture analysis was performed (Figure 2). Since contrast and brightness variations can affect the true texture of the image, a gray-level normalization for each ROI was performed by using the limitation of dynamics to μ ± 3σ (μ = gray-level mean; and σ = gray-level standard deviation) [23].

The features computed by the software’ generator originated from the gray-level histogram, the absolute gradient, the wavelet transformation, the co-occurrence matrix, the run-length matrix, and the autoregressive model. The features generated by MaZda software include several hundred elements per ROI. Since such a large number of data is hard to be handled by common statistical analysis software, the MaZda software allows the selection of the most discriminative features through reduction techniques. These techniques highlight subsets of features that allow minimum error classification of analyzed image textures [24]. One such technique is represented by the Fisher selection method. The Fisher coefficient (F) defines the ratio of between-class variances to within-class variances [25]. Overall, this method provides a set of 10 features that have a high discriminatory ability. Alongside the Fisher method, another selection technique based on the probability of classification error and average correlation coefficients (POE + ACC) was utilized [24]. By applying these selection methods, two sets, each containing 10 features, were selected. The first one contained highly discriminative features produced by the Fisher method, and the second one was based on the minimization of classification error generated by the POE + ACC technique.

To evaluate texture features and individual ability to discriminate within groups, their absolute values were compared using the independent samples *t*-test. The receiver operating characteristic (ROC) analysis was performed, with the calculation of the area under the curve (AUC) with 95% confidence intervals (CIs) for the parameters showing *p*-values below 0.05 on the univariate analysis. Optimal cut-off values were chosen using an optimization step that maximized the Youden index for predicting patients with HGGs, and sensitivity and specificity were computed from the same data, without other adjustments. The ROC curves’ comparisons were conducted using the DeLong et al. method [26]. Secondly, a multiple regression analysis was performed, using an “enter” input model (which consisted of inputting all variables in the model in one single step), to identify which of the texture parameters that showed statistically significant results at the univariate analysis are also independent predictors of HGGs. The coefficient of determination (R-squared) was computed, and the diagnostic value of the prediction model was evaluated using ROC analysis. Statistical analysis was performed using a commercially available dedicated software MedCalc version 14.8.1 (MedCalc Software, Mariakerke, Belgium).

The feature name generated by MaZda software contains abbreviations of feature characteristics produced by the extraction algorithm. The outermost symbol from the left indicates the first imaging processing procedure. The first letter indicates the color channel (“C” implies that a black and gray image was computed, “R” identifies the red color channel). The second symbol stands for image normalization (N) after which comes the encoding for the method used, in this case “S”, which represents image normalization using the limitation of dynamics to μ ± 3σ. The following number indicates the feature was quantized to use that particular number of bits per pixel. The direction is coded using letters: H (horizontal), V (vertical), Z (45°), and N (135°). The next group of letters identifies the extraction algorithm (e.g., Wav, Haar wavelet transformation). The feature name is usually the last group of letters (LngREmph, long-run emphasis; ShrtREmp, short-run emphasis; En, energy) [27]. The MaZda software was also used to generate maps that show the distribution of a particular texture parameter in computed images.

## 3. Results

Most texture features selected by the Fisher method derived from the first-order histogram and run-length matrix, while the ones highlighted by the POE + ACC method originated from the co-occurrence matrix and wavelet transformation. Four texture parameters (1% percentile (Perc01), 10% percentile (Perc10), wavelet energy (WavEnLL_s-4), and the fraction of image in runs (RNS6Fraction)) were highlighted following both selection methods. The comparison of HGGs and BMs based on the absolute values of the parameters selected by the Fisher method held statistically significant results in all 10 cases. The comparison between the two entities based on POE + ACC set reached statistically significant results for five parameters, of which four were the same previously selected by the Fisher method. The feature sets selected by Fisher and POE + ACC methods along with the univariate analysis results are displayed in Table 1. Overall, the parameters showing statistically significant results derived from histogram analysis, wavelet transformation, and the run-length matrix (Table 2).

The ROC analysis results are displayed in Table 3. Considering all calculated parameters, Perc01 yielded the highest AUC (0.858; CI, 0.716–0.946) which was statistically different from all parameters derived from histogram analysis, but from the ones computed by wavelet transformation and run-length matrix (Table 4). Overall, the first-order histogram-derived parameters obtain a high specificity at a cost of a medium-to-high sensibility for the differentiation of HGGs from BMs, while parameters computed from wavelet transformation and rung-length matrix showed opposite results. The *p*-values for the ROC curve comparisons are reported in Table 4. The ROC curves of the six parameters that showed the highest AUCs are displayed in Figure 3.

The multiple regression analysis that integrated all parameters showing statistically significant results at the univariate analysis showed that Perc01 (*p* = 0.037) and WavEnLL_s-4 (*p* = 0.031) were independent predictors of high-grade gliomas (Table 5). The variance inflation factor (VIF) yielded high values for most texture parameters, which indicates the multicollinearity of the independent variables. The mean parameter was excluded from the prediction model due to the high VIF values (VIF>10^4^). The overall prediction model showed an approximately equal sensitivity (100%; CI, 29.2–100%) and specificity (66.7%; CI, 49.8–80.9%), and a highest AUC (0.964; CI, 0855–0.997) for the diagnosis of HGGs than the ones exhibited by the parameters included in the model. The ROC comparison between the prediction model and the two independent predictors showed statistically significant results (Perc01, *p* = 0.04; WavEnLL_s-4, *p* = 0.003; Figure 2B).

## 4. Discussion

Our results show that 16 individual texture parameters were highlighted by the selection methods, with 11 recorded statistically significant results at the univariate analysis. Perc01 showed the highest AUC (0.858; CI, 0.716–0.946), while also being an independent predictor of HGGs, together with the WavEnLL_s-4 parameter.

Because the edemas encountered in HGGs and BMs express high-intensity signals on both T2W and FLAIR sequences, it is very difficult for them to be macroscopically differentiated through conventional MRI examinations [1]. Mucio and colleagues [28] observed a moderate diagnostic value when differentiating between the two based on signal alteration in the adjacent cortex (sensitivity, 60.7%; specificity, 67.9%). Additionally, the results of functional MRI diffusion-weighted sequences show contradictions. Some studies [10,29,30] found a restricted water diffusion along the margins of HGGs and attributed it to malignant cell infiltration, while others [28,31,32] concluded that there is no difference in diffusion measurements of the peritumoral edema between the two entities. More advanced MRI techniques such as multivoxel proton magnetic resonance spectroscopy have also been used to assess the two types of peritumoral regions, with uncertain outcomes. Tsugos and colleagues [33] found a decrease in N-acetyl-aspartate/Creatinine ratio in the peritumoral areas of glioblastomas compared to BMs, Jiun-Lin Yan et al. [34] found no statistically significant difference between the two, and Wijnen et al. [35] reported an increase of this ratio. Perfusion MRI techniques such as dynamic susceptibility contrast MRI can also be used to quantify the peritumoral infiltration of glioblastomas due to their increased perfusion and angiogenesis [12]. The high values of the relative cerebral blood volume (rCBV) in the peritumoral area were found to be strongly correlated with cellular proliferation of HGGs by some authors [13,34], while others [36] found it to be a poor diagnostic feature. In terms of fractional anisotropy, Bette et al. [37] evaluated 294 patients with HGGs and BMs and found no significant differences in the peritumoral area.

Most texture analysis studies involving gliomas focus on intratumoral characteristics, while the surrounding area of these lesions remains relatively unexplored. Differentiating high-grade from low-grade gliomas is one of the most researched topics in this domain [20,21,38,39,40]. Despite variability in image segmentation and TA software, the entropy parameter computed from apparent diffusion coefficient (ADC) maps consistently showed very good results in differentiating the two [20,39]. We were able to find only a few studies compatible with our research. Skogen et al. [41] investigated the utility of TA in the differentiation of glioblastomas from brain metastases based on the peritumoral area as seen on diffusion tensor images (DTI). Although a similar number of patients were enrolled (*n* = 43), the workflow was almost entirely different. TA was performed on a commercially available software (TexRAD) which provided the analysis of five texture parameters (mean, standard deviation, entropy, skewness, and kurtosis) and the ROIs were freehand-drawn within 1 cm of the tumor boundaries of fractional anisotropy (FA) and ADC maps. Entropy was again the parameter with the highest discriminatory ability following its computation from FA maps (80% sensitivity, 90% specificity, and an AUC of 0.88) followed by the standard deviation [41]. Furthermore, TA of the peritumoral area was also performed by Mouthuy et al., [42] who concluded that by combining texture features with rCBV parameters, the differentiation of two entities has an increased performance (92% sensitivity, 71% specificity). The good results of the peritumoral area assessment using TA were also observed by Artzi et al., [43], which by the use of a machine-learning algorithm (support-vector-machine) accomplished an overall 87% accuracy, 86% sensitivity, and 89% specificity for the training test data in distinguishing the two entities.

The first-order histogram is one of the most common statistical methods for image feature computation. The histogram does not consider the spatial relations between the pixels, reflecting only the value of their intensity [44] through parameters such as mean, standard deviation, variance, skewness, kurtosis, and percentiles. Out of nine histogram features that can be generated by MaZda [22], six were highlighted by the selection methods and showed statistically significant results when comparing HGGs and BMs. In all six cases, the average values were higher for BMs than for HGGs. The mean parameter reflects the average value of the pixels within the ROI [45]. The percentile number (n) is the point at which n% of the pixel values that form the histogram are found to the left [46]. In other words, a percentile gives the highest gray-level value under which a given percentage of the pixels in the image are contained [47]. This signifies that 1%, 10%, 50%, 90%, and 99% of the pixels within images were distributed under lower values for HGGs than for BMs, and the average pixel intensity was lower for the primary tumors. It is possible that the higher pixel values recorded in the peritumoral zone of BMs could be a result of the pure vasogenic edema that surrounds these lesions, the increase extracellular water due to the breakdown of the blood–brain barrier could lead to an increase in the T2W signal. Previous studies have also successfully demonstrated the role of first-order histogram percentiles in the diagnosis of glial tumors. Two studies [20,48] using texture features computed from ADC maps showed that the 5th percentile derived from the whole-tumor ROI successfully differentiated between high- and low-grade gliomas, while also being strongly correlated with the Ki-67 labeling index (*p* = 0.003) [48]. Additionally, in our study, one of the highest AUCs was obtained by two percentiles (Perc01, AUC = 0.858; Perc50, AUC = 0.772) which indicates that they possess a high practical value for the diagnosis of gliomas.

The gray-level run-length matrix (RLM) is a method of extracting higher-order statistical texture features [49]. The RLM quantifies pixel runs with a specific grayscale level and length [27]. A run is a line of pixels having the same intensity values. There are four directions of pixel runs. The number of pixels within a run defines the run length, and the value of the run length is given by the number of occurrences [49]. Three parameters derived from the RLM, all computed for the 135° direction, showed statistically significant results when comparing HGGs with BMs: short-run emphasis (RNS6ShrtREmp, *p* = 0.0045), long-run emphasis (RNS6LngREmph, *p* = 0.0083), and the fraction of image in runs (RNS6Fraction, *p* = 0.0057). Short- and long-run emphasis reflects the distribution of short or long homogeneous runs in an image [50]. Higher values of short-run emphasis indicate fine textures, while higher values of long-run emphasis reflect coarse surfaces [51]. The percentage of pixels that are part of any of the runs is measured through the fraction of images in runs [47]. The latter reflects the ration of the total number of runs in the image to the total number of pixels in the image and it is not relevant from a histopathological or imaging point of view in this case. We obtained higher average values of short-run emphasis for BMs and higher average long-run emphasis values for HGGs. These reflect a rougher peritumoral zone in the case of gliomas. This could reflect the difference between neoplastic invasion and vasogenic edema, the infiltrative growth of HGGs causing irregularities in pixel intensities, and breaking the pixel runs. The maps that show the distribution of short- and long-run emphasis in images of HGGs and BMs are shown in Figure 4.

Wavelet transformation is a multiresolution technique that aims to transform images into a representation that can contain spatial as well as frequency information [52]. This transformation allows the quantification of the frequency content of an image, which is directly proportional to the gray-level variations within that image. Firstly, images are scaled up five times both in vertical and horizontal directions. Furthermore, two types of filters (high and low pass) are applied to separate the image data [53]. Finally, different subbands are extracted from an image that becomes subdivided into four parts, corresponding to different frequency components. The result is a five-scaled image with four frequency bands on every scale, each labeled as low–low (LL), high–low (HL), low–high (LH), and high–high (HH). The LL band contains low-frequency signal contents, whereas the HH band contains high-frequency signal contents, which carry less importance than the LL band [52]. A five-level decomposition diagram of a T2W image of glioblastoma is displayed in Figure 5. The energy feature can be computed from each subband [52]. Wavelet energy quantifies the distribution of energy along the frequency axis over scale and orientation [54]. Energy measures the local uniformity within an image. When the gray levels of an image are distributed under a constant or periodical form, energy becomes high. Contrary, the value of this parameter decreases when multiple entries are present within the image matrix. [24]. Two wavelet-based energy features showed statistical significance when comparing HGGs with BMs: energy computed from the low–low frequency band within the forth image scale (WavEnLL_s-4, *p* = 0.0062) and energy computed from the high–high frequency band within the first image scale (WavEnHH_s-1, *p* = 0.0067). The first parameter (WavEnLL_s-4) showed higher values for HGGs than for BMs (10,272.3 ± 4385.8 versus 6579.94 ± 2732.81), while in the case of the second feature (WavEnHH_s-1), the values were reversed (6.25 ± 3.47 for HGGs versus 10.96 ± 7.11 for BM). Since the LL band frequency carry less important textural information than HH [24], we consider WavEnLL_s-4 values as a predictor of irregularity in the peritumoral zone where gliomas neoplastic infiltration determines nonuniform pixel intensity variations within the edema [41].

In our study, we demonstrated via texture analysis that the peritumoral edema of HGGs is more heterogeneous and contains higher pixel intensity variations compared to BMs. Additionally, the peritumoral zone of BMs seems to exhibit higher signal intensities on T2W images, probably due to the absence of contamination with tumoral cells. These observations are compatible with previous research. Skogen et al. [41] concluded that the peritumoral edema of glioblastomas expresses a high degree of irregularity on FA and ADC maps, as quantified through the texture parameters entropy and standard deviation of pixel intensity. However, we consider that the use of a classic T2W sequence is more approachable since it is routinely performed on every MRI brain examination and has fewer limitations than DTI [55]. Additionally, it is possible that the TA characteristics of any examined tissue are better highlighted as there are more computed TA features, as they can offer a more complete and complex characterization of the regions of interest.

Signal-to-noise ratio, spatial resolution, magnetic strength as well as other acquisition parameters can have an important influence on TA results [56]. We counteracted such effects through ROI normalization but also by selecting only examinations performed by the same protocol and on the same machine, thus providing a high degree of homogeneity in the current study. Supra-tentorial resection of gliomas is guided by the tumoral margins, visible on the FLAIR sequence because it can provide superior delineation of the white matter lesions [28,57]. The only issue is that the spatial orientation of the FLAIR sequence was volatile in the included examinations (alternating between axial and coronal). To preserve the homogeneity of the lot, a T2W sequence was used that was the same in all studies. The strict selection of supratentorial lesions together with the above-mentioned procedures allowed us to counteract the variations of the textural measurements produced by the use of multiple scanners or different examination protocols [27,58].

When investigating through different modalities (such as spectroscopy or diffusion tensor imaging) the MRI characteristics of peritumoral zones of gliomas, most researchers used a freehand ROI to cover these regions [59,60]. Although such modalities are dependent on the examiners’ experience, they produce inconsistencies between the measurements made even by the same researcher [61]. Besides, it is important to consider that the peritumoral cellular infiltration gradually decreases from the tumor edges towards the periphery [62] and can be present even in areas with normal T2 signals [61]. For this reason, it might be possible that a semiautomatic or fully automated technique based on a regional gradient would better encompass the extension of this area.

Advanced MRI techniques such as spectroscopy [13,33,34], perfusion MRI [12], and DTI [41] were shown to be useful in assessing both the tumor and the peritumoral region of gliomas. However, rather than performing advanced techniques and adding on sequences, it is important to quantify and utilize information from images already obtained, as they can carry additional diagnostic information. In this regard, we chose to assess texture information from a basic MRI sequence since it is routinely used in brain examination protocols. However, despite the promising results, texture analysis (and particularly histogram analysis) is not incorporated in most guidelines [63,64]. To enter clinical practice, TA techniques require large prospective multicentric studies and adequate software to be incorporated in workstations.

Our study has several limitations. Firstly, the number of BMs included exceeded one of the HGGs by about 60%, and the overall number of lesions was relatively small. Due to the organization of our healthcare unit, we preferably perform the postsurgical follow-up examinations, while the preoperative assessment is mostly performed in another sector. Secondly, the lack of direct correlation of texture parameters with histological findings in the peritumoral zone can also be viewed as a limitation. Furthermore, owing to its retrospective design, it may have selection bias. Thirdly, a multivariate analysis (such as logistic regression) was also demonstrated to be a useful modality to select discriminative texture features in previous studies [65,66]. To build a robust predictive model, the feature subset should contain mostly uncorrelated parameters [66]. Therefore, our selection method involving Fisher coefficients highlights parameters that besides having a high discriminatory potential, are also well correlated with each other [67]. Thus, this aspect may have affected the overall diagnostic value of our combined prediction model. Additionally, the ROI segmentation employed in this pilot study comprised a single largest cross-section-based delineation instead of a multislice or three-dimensional volume analysis. However, previous studies using the filtration-histogram technique have demonstrated the comparison of single-slice vs. multislice/volume analysis on computer tomography (CT) in primary colorectal cancer for prognostication [68] as well as on MRI in gliomas for IDH versus wild-type differentiation [69]. Interestingly the analysis demonstrated single-slice analysis was significant in predicting prognosis in colorectal cancer on CT [68] and IDH vs. wild-type differentiation in gliomas on MRI [69] and comparable to multislice/volume analysis. It is not therefore clear if there is any “significant” added-value of undertaking multislice/volumetric analysis which not only entails increased analysis time (barrier to adoption in a busy-clinic) and increased operator variability associated with multislice/volume analysis. Another limitation can potentially come from the fact that no intra- or interobserver agreement was assessed. However, previous studies on CT and MRI have demonstrated good reproducibility for filtration-histogram based TA using multicenter clinical validation [70,71], robustness to variation in image acquisition parameters [71,72,73], and good inter- and intraoperator repeatability (good intraclass correlation from test-retest technique) [74,75]. Additionally, previous research following the same method stated that due to the semiautomatic ROI positioning, this assessment is not necessary [76]. Additionally, the MaZda software used in this article can be regarded as outdated, since the official version had not received improvements in more than 10 years. However, in this study, we used a newly developed Beta version of this software, released in 2016 (Available online: https://data.mendeley.com/datasets/dkxyrzwpzs/1). Although modern dedicated TA software is both free and commercially available, MaZda steel represents a valid TA method, since it provides one of the largest numbers of feature customization, selection, extraction, and processing methods. Additionally, it offers an intuitive interface, and thus the possibility of being used by nonimage processing specialists, such as regular physicians. The MaZda software also enables the use of several classifiers. In this paper, we preferably applied a more conventional approach to the statistical processing of the parameters, since several of these classifiers (such as the artificial neural networks) require big data for an adequate classification procedure.

## 5. Conclusions

In conclusion, our study showed that high-grade gliomas and solitary brain metastases can be successfully differentiated based on the textural information extracted from the peritumoral zone. Features derived from the histogram analysis, wavelet transformation, and run-length matrix show high discriminatory potential both as stand-alone parameters as well as incorporated in a predictive model. The results of this study support the hypothesis that the textures of the peritumoral edema can reflect the neoplastic cell infiltration of high-grade gliomas, but further research is required to validate this method.

## Figures and Tables

**Figure 1 brainsci-10-00638-f001:**
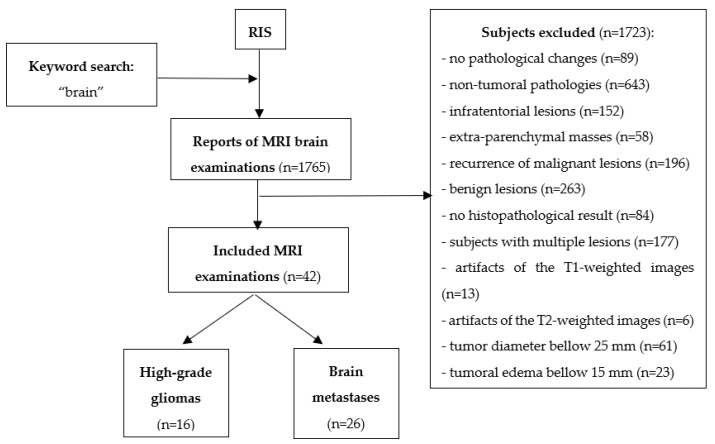
Patients. RIS, radiology information system; MRI, magnetic resonance imaging.

**Figure 2 brainsci-10-00638-f002:**
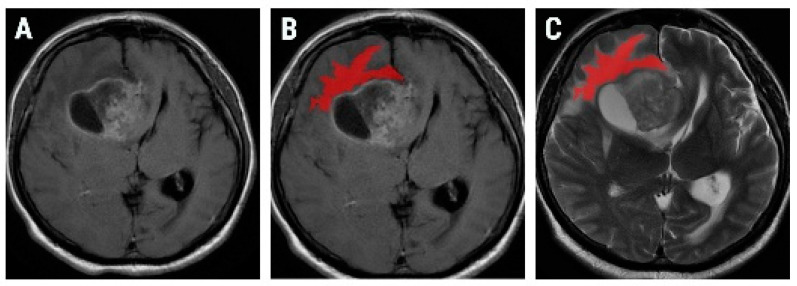
Axial contrast-enhanced T1-weighted image of a 56-year-old patient with pathologically proven glioblastoma (**A**) and the region of interest (red) overlapping the peritumoral area (**B**) on a postcontrast T1-weighted image, which was consequentially transferred on to a synchronized slice on the T2-weighted sequence (**C**).

**Figure 3 brainsci-10-00638-f003:**
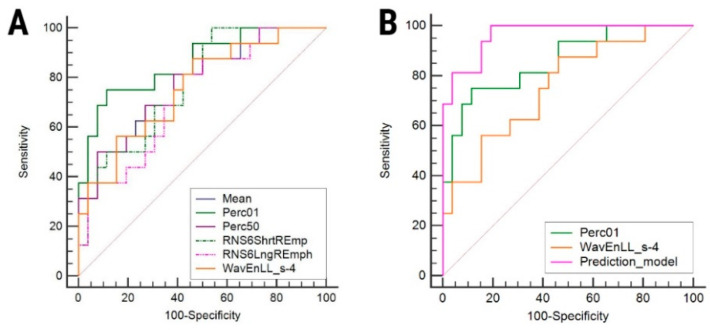
Comparison of receiver operating characteristic (ROC) curves between (**A**) the six texture parameters that showed the highest area under the curve, and (**B**) independent parameters and the predictive model for the diagnosis of high-grade gliomas. Mean, histogram mean; Perc01/50, 1%/50% percentile; ShrtREmp, short-run emphasis; LngREmph, long-run emphasis; WavEn, wavelet energy.

**Figure 4 brainsci-10-00638-f004:**
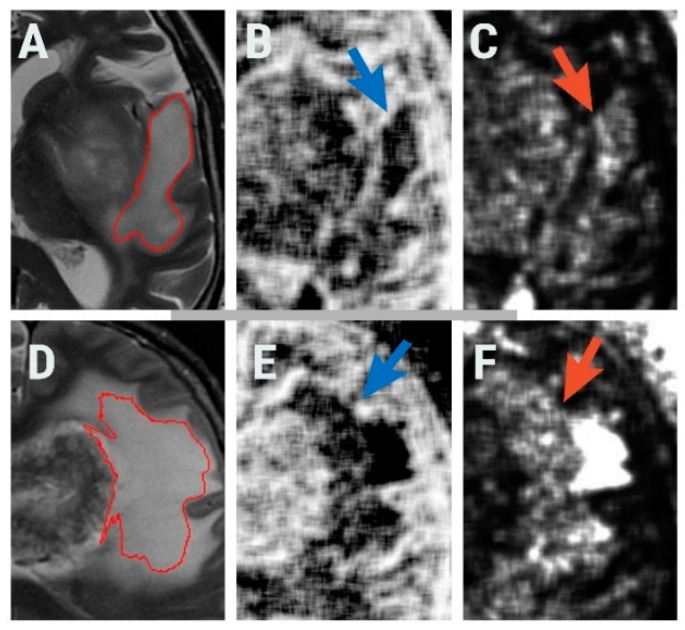
Axial T2-weighted image of a 61-year-old patient with glioblastoma (**A**) and the region of interest (red) used for texture analysis; (**B**) generated map based on short-run emphasis parameter (blue arrow pointing to the peritumoral zone); (**C**) generated map based on long-run emphasis parameter (orange arrow pointing to the peritumoral zone; axial T2-weighted image of a 68-year-old patient with brain metastases (**D**) and the region of interest (red) used for texture analysis; (**E**) generated map based on short-run emphasis parameter (blue arrow pointing to the peritumoral zone); (**F**) generated map based on long-run emphasis parameter (orange arrow pointing to the peritumoral zone.

**Figure 5 brainsci-10-00638-f005:**
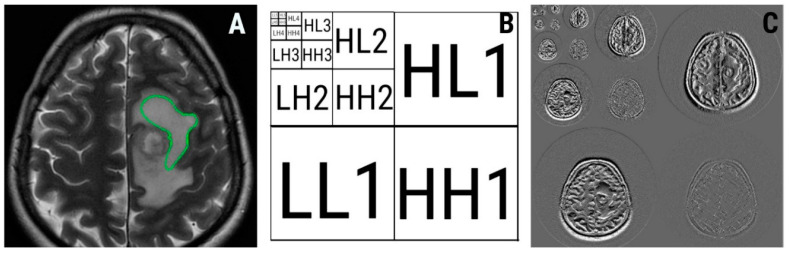
Axial T2-weighted image of a 72-year-old patient with glioblastoma (**A**) and the region of interest (green) used for texture analysis; (**B**) five-level wavelet decomposition diagram; (**C**) five-level wavelet decomposition of (**A**). The numbers represent the decomposition levels. Frequency bands are noted: LL, low–low; HL, high–low; LH, low–high; HH, high–high.

**Table 1 brainsci-10-00638-t001:** Sets of features generated by each selection method and the univariate analysis results following the comparison of high-grade gliomas with brain metastases.

Fisher	F	*p*-Value	POE + ACC	PP	*p*-Value
Perc01 *	2.31	**<0.0001**	CV5S6SumOfSqs	0.39	0.0883
Perc10 *	1.73	**0.0002**	CV4S6InvDfMom	0.41	0.0997
Mean	1.27	**0.0013**	WavEnHH_s-1	0.46	**0.0067**
Perc50	1.22	**0.0015**	WavEnHL_s-4	0.47	0.3853
WavEnLL_s-4 *	1.2	**0.0062**	Perc10 *	0.47	**0.0002**
RNS6ShrtREmp	0.96	**0.0045**	RNS6Fraction *	0.49	**0.0057**
Perc90	0.92	**0.0053**	WavEnLL_s-4 *	0.49	**0.0062**
RNS6Fraction *	0.9	**0.0057**	CZ5S6SumAverg	0.49	0.5007
RNS6LngREmph	0.81	**0.0083**	WavEnLH_s-4	0.49	0.7559
Perc99	0.78	**0.0096**	Perc01 *	0.64	**<0.0001**

* parameters highlighted by both classification methods. Bold values are statistically significant. F, Fisher coefficients; POE + ACC, probability of classification error and average correlation; PP, POE + ACC coefficients; *p*-value showing the univariate analysis result; Perc 01/10/50/90/99, 1%/10%/50%/90%/99% percentile; Mean, histogram mean; WavEn, wavelet energy; ShrtREmp, short-run emphasis; Fraction, the fraction of image in runs; LngREmph, long-run emphasis; SumOfSqs, the sum of squares; InvDfMom, inverse difference moment; SumAverg, sum average.

**Table 2 brainsci-10-00638-t002:** The parameters that show statistically significant results at the univariate analysis and their average values recorded in each group.

Parameter	HGGs	BMs
Perc01	33,848.43 ±328.15	34,308.65 ± 298.8
Perc10	33,994.5 ± 363.17	34,437 ± 322.34
Perc50	34,182.12 ± 433.34	34,581.69 ± 325.32
Perc90	34,331.18 ± 466.79	34,699.46 ± 341.39
Perc99	34,411.31 ± 489.28	34,765.03 ± 352.9
Mean	34,171.97 ± 420.92	34,573.13 ± 325.66
WavEnLL_s-4	10,272.3 ± 4385.84	6579.94 ± 2732.81
WavEnHH_s-1	6.25 ± 3.47	10.96 ± 7.11
RNS6Fraction	0.9 ± 0.02	0.93 ± 0.02
RNS6ShrtREmp	0.93 ± 0.01	0.94 ± 0.01
RNS6LngREmph	1.32 ± 0.11	1.23 ± 0.08

Data are expressed as mean ± standard deviation. HGGs, high-grade gliomas; BMs, brain metastases; Perc 01/10/50/90/99, 1%/10%/50%/90%/99% percentile; Mean, histogram mean; WavEn, wavelet energy; ShrtREmp, short-run emphasis; Fraction, the fraction of image in runs; LngREmph, long-run emphasis.

**Table 3 brainsci-10-00638-t003:** Receiver operating characteristic (ROC) analysis results of the texture parameters in high-grade gliomas’ assessment.

Parameter	Sign. Lvl.	AUC	J	Cut-Off	Sensitivity (%)	Specificity (%)
Perc01	<0.0001	0.858 (0.716–0.946)	0.63	≤34,039	75 (47.6–92.7)	88.46 (69.8–97.6)
Perc10	0.0031	0.748 (0.59–0.869)	0.53	≤34,081	68.75 (41.3–89)	84.62 (65.1–95.6)
Perc50	0.0003	0.772 (0.616–0.887)	0.42	≤34,466	81.25 (54.4–96)	61.54 (40.6–79.8)
Perc90	0.006	0.726 (0.567–0.852)	0.37	≤34,728	87.5 (61.7–98.4)	87.5 (61.7–98.4)
Perc99	0.0084	0.719 (0.559–0.846)	0.37	≤34,831	87.5 (61.7–98.4)	87.5 (61.7–98.4)
Mean	0.0002	0.774 (0.619–0.889)	0.42	≤34,154.86	50 (24.7–75.3)	92.31 (74.9–99.1)
WavEnLL_s-4	0.0009	0.757 (0.6–0.876)	0.41	>6458.11	87.5 (61.7–98.4)	53.85 (33.4–73.4)
WavEnHH_s-1	0.0294	0.68 (0.519–0.816)	0.38	≤14.8	100 (79.4–100)	38.46 (20.2–59.4)
RNS6Fraction	0.0004	0.748 (0.606–0.88)	0.46	≤0.94	100 (79.4–100)	46.15 (22.6–66.6)
RNS6ShrtREmp	0.0001	0.776 (0.622–0.89)	0.46	≤0.95	100 (79.4–100)	46.15 (22.6–66.6)
RNS6LngREmph	0.0041	0.728 (0.596–0.854)	0.38	>1.23	81.25 (54.4–96)	57.69 (36.9–76.6)

Between brackets are the values corresponding to the 95% confidence interval. Sign.lvl., significance level; J, Youden index; Perc 01/10/50/90/99, 1%/10%/50%/90%/99% percentile; Mean, histogram mean; WavEn, wavelet energy; ShrtREmp, short-run emphasis; Fraction, the fraction of image in runs; LngREmph, long-run emphasis; AUC, area under the curve.

**Table 4 brainsci-10-00638-t004:** Comparison of ROC curves in the differentiation of high-grade gliomas from brain metastases. Numbers represent *p*-values. Each *p*-value column represents the comparison between all parameters and the reference one (REF). Values in bold are statistically significant.

**Perc01**	**REF**	**0.0021**	**0.0362**	**0.0109**	**0.0095**	**0.0231**	0.2766	0.0572	0.2933	0.3627	0.17
**Perc10**	**0.0021**	REF	0.5385	0.6518	0.5633	0.4581	0.9311	0.552	0.9007	0.7992	0.872
**Perc50**	**0.0362**	0.5385	REF	**0.0203**	**0.0264**	0.7778	0.8862	0.3996	0.7568	0.964	0.7041
**Perc90**	**0.0109**	0.6518	**0.0203**	REF	0.3795	**0.0233**	0.7691	0.6882	0.7568	0.6589	0.9843
**Perc99**	**0.0095**	0.5633	**0.0264**	0.3795	REF	**0.0256**	0.7191	0.7373	0.7122	0.6169	0.9377
**Mean**	**0.0231**	0.4581	0.7778	**0.0233**	**0.0256**	REF	0.8668	0.1105	0.9103	0.9817	0.6822
**WavEnLL_s-4**	0.2766	0.9311	0.8862	0.7691	0.7191	0.8668	REF	0.4324	0.9553	0.8246	0.7434
**WavEnHH_s-1**	0.0572	0.552	0.3996	0.6882	0.7373	0.1105	0.4324	REF	0.1105	0.0655	0.3928
**RNS6Fraction**	0.2933	0.9007	0.7568	0.7568	0.7122	0.9103	0.9553	0.1105	REF	0.358	0.0693
**RNS6ShrtREmp**	0.3627	0.7992	0.964	0.6589	0.6169	0.9817	0.8246	0.0655	0.358	REF	0.0705
**RNS6LngREmph**	0.17	0.872	0.7041	0.9843	0.9377	0.6822	0.7434	0.3928	0.0693	0.0705	REF

Perc 01/10/50/90/99, 1%/10%/50%/90%/99% percentile; Mean, histogram mean; WavEn, wavelet energy; ShrtREmp, short-run emphasis; Fraction, the fraction of image in runs; LngREmph, long-run emphasis.

**Table 5 brainsci-10-00638-t005:** Multivariate analysis of factors independently associated with the presence of high-grade gliomas. Bold values are statistically significant.

Independent Variable	Coefficient	Standard Error	*p*-Value	VIF
Perc01	−0.002	0.001	**0.0370**	67.869
Perc10	0.001	0.002	0.4061	247.596
Perc50	0.0008	0.003	0.7806	555.997
Perc90	−0.006	0.006	0.3198	2433.857
Perc99	0.005	0.004	0.2162	1245.022
RNS6Fraction	0.91	77.72	0.9906	1224.984
RNS6LngREmph	3.08	10.38	0.7682	405.915
RNS6ShrtREmp	−19.75	49.49	0.6925	270.604
WavEnHH_s-1	−0.004157	0.01405	0.7692	2.693
WavEnLL_s-4	0.0000413	0.0000182	**0.0311**	1.672
Sign. level.	**0.0002**			
R^2^	0.6180			
R^2^ adjusted	0.4948			
M.R. Coef.	0.7861			

VIF, variance inflation factor; R^2^, coefficient of determination; R^2^ adjusted, coefficient of determination adjusted for the number of independent variables in the regression model; Sign. level, the significance level of the multivariate analysis; M.R. Coef., multiple correlation coefficient.

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
