# Peer review of "Differentiating High-Grade Gliomas from Brain Metastases at Magnetic Resonance: The Role of Texture Analysis of the Peritumoral Zone"

_brainsci, 2020, doi:10.3390/brainsci10090638_

Round 1

Reviewer 1 Report

This excellent work establishes a software-based texture analysis algorithm for differentiating high-grade gliomas (HGGs) from brain metastases (BMs). This algorithm differentiate with a relatively high sensitivity and specificity between HGGs and BMs. All results are sound and the conclusions drawn are convinving. The linguistic style is very good, making the manuscript easy to read and understand. The results presented could be of high practical value for patient care.

In summary, I strongly recommend publication without further changes needed.

Author Response

Thank you very much for the time and effort you invested in reading, analyzing, and reviewing our manuscript. Your remarks are deeply appreciated.

Reviewer 2 Report

The main objection to work is the number of cases. The statistical analysis is based on too many parameters. Statisticians recommend to use 10 cases per one parameter. If you don't have more cases then you should add more ROIs on different slices to the analysis. 

Also tumor zone TA analysis would benefit from new informations. 

The discussion is worthless when the statistics fail.

 Other comments :

1. In the introduction, only one sentence takes into account the differentiation between HGG and BT (lines 72-74) It is your main topic so it should be much longer. 

2. In the introduction only three sentences focus on texture analysis. Please introduce this technique and analysed parameters. Tell the readers what these parameters mean and what they represent in images.

3. Please add the diagram showing the procedure for selecting patients. 

4. Was the T2-weighted sequence performed by TSE technique or another? Which acquisition technique was used to acquire the T1w?

5. Line 163 - DeLong - reference is missing

6. Line 164 "" enter" input model" - What is this model? The term is unclear. 

7. The summary of the results should be in first paragraph of the discussion

8. No consistent reference of the results to the literature in the topic

9. The article lacks information on TA parameters when differentating the entire tumor fraction and comparing it to the parameters when analyzing only the edema 

10. The discussion is so extensive that it is difficult to follow the idea of the article, it is worth shortening it. 

Reviewer 3 Report

In this manuscript, the authors have developed a texture analysis model for differentiating high-grade gliomas from brain Metastases at magnetic resonance. This model was applied to a retrospective brain examination study  performed at University of Medicine and Pharmacy Cluj-Napoca between May 2016 and March 2020. This a well designed study and the model has been applied quite judiciously to this retrospective study. Kindly find my comments below:

Major Comments

  1. How did the authors validate the developed texture analysis model? Did the authors employ training set and validation set to the data to measure the accuracy of the model in differentiating high-grade gliomas from brain Metastases.
  2. Since this is a retrospective study design, how was the heterogeneity in data addressed?
  3. Could the authors provide detailed demographics and clinical pathologies of the patients selected for the current study ? 

Reviewer 4 Report

As author described, there are many limitations in this study.

However, author reported new findings with appropriate analysis.

Author Response

(The authors gave the same response as above.)
